

# Technical note: Boundary layer height determination from Lidar for improving air pollution episode modelling: development of new algorithm and evaluation

Ting Yang[1*,] Zifa Wang[1*], Wei Zhang[2], Alex Gbaguidi[1], Nobuo Sugimoto[3], Xiquan Wang[1], Ichiro Matsui[3], Yele Sun[1]

[1] State Key Laboratory of Atmospheric Boundary Layer Physics and Atmospheric Chemistry, Institute of Atmospheric Physics, Chinese Academy of Sciences, Beijing 100029, China.
[2] Aviation Meteorological Center of China, Beijing 100021, China
[3] National Institute for Environmental Studies, 16-2 Onogawa, Tsukuba, 305-8506, Japan

*Correspondence to: Zifa Wang (zifawang@mail.iap.ac.cn), Ting Yang (tingyang@mail.iap.ac.cn)*

**Abstract.** Predicting air pollution events in low atmosphere over megacities requires thorough understanding of the tropospheric dynamic and chemical processes, involving notably, continuous and accurate determination of the boundary layer height (BLH). Through intensive observations experimented over Beijing (China), and an exhaustive evaluation existing algorithms applied to the BLH determination, persistent critical limitations are noticed, in particular over polluted episodes. Basically, under weak thermal convection with high aerosol loading, none of the retrieval algorithms is able to fully capture the diurnal cycle of the BLH due to pollutant insufficient vertical mixing in the boundary layer associated with the impact of gravity waves on the tropospheric structure. Subsequently, a new approach based on gravity wave theory (the cubic root gradient method: CRGM), is developed to overcome such weakness and accurately reproduce the fluctuations of the BLH under various atmospheric pollution conditions. Comprehensive evaluation of CRGM highlights its high performance in determining BLH from Lidar. In comparison with the existing retrieval algorithms, the CRGM potentially reduces related computational uncertainties and errors from BLH determination (strong increase of correlation coefficient from 0.44 to 0.91 and significant decrease of the root mean square error from 643 m to 142 m). Such newly developed technique is undoubtedly expected to contribute to improve the accuracy of air quality modelling and forecasting systems.

## 1 Introduction

The boundary layer height (BLH) illustrates the relationships between air pollution intensity, duration and scope; it constitutes an important factor influencing the diffusion of pollutants in low atmosphere (Tie et al., 2007;Quan et al., 2013). An increase of air pollutants is often associated with a shallow BLH, while a decrease of pollutants is accompanied by obvious uplift of the BLH. Besides the physical effects, BLH can also affect the precursor particles concentration and distribution, which might affect the chemical transformation of fine particulate matter (Ansari and Pandis, 1998). BLH is also a key parameter for air pollution models; it determines the volume available for the dispersion of pollutants and is involved in many predictive and diagnostic methods and/or models to assess pollutant concentrations (Seibert et al., 2000). The bias of the BLH between the air quality model and observation is a potential cause of model's difficulties to



accurately forecast air pollution episodes (Dabberdt et al., 2004). Therefore, accurately acquiring the BLH, especially during polluted episodes, is of great significance to investigate air pollution issues.

Many techniques have been developed to determine the BLH, e.g., through radiosonde measurements (Stull, 1988), remote sensing (Emeis et al., 2007), laboratory experiments (Park et al., 2001) and model simulations (Dandou et al.,
2009). The high spatio-temporal resolutions make aerosol Lidar techniques (light detection and ranging)  as one of the most suitable systems for analyzing the boundary layer structure and determining the BLH (Flamant et al., 1997). Due to the complex vertical structure of boundary layer,  numerous methods have been proposed to accurately retrieve the BLH from Lidar, such as maximum variance method (Hooper and Eloranta, 1986), fitting idealized profile method (Steyn et al., 1999), first point method (Boers and Melfi, 1987), threshold method  (Dupont et al., 1994), wavelet transform method
(Davis et al., 2000;Baars et al., 2008), first gradient method (Flamant et al., 1997), logarithm gradient method (Senff et al., 1996), and normalized gradient method (He et al., 2006). However, most of the algorithms have been tested and validated only over relatively unperturbed homogeneous terrain, e.g., oceans (Melfi et al., 1985;Flamant et al., 1997), rural areas, and clean meteorological conditions (Piironen and Eloranta, 1995) so far. Limited evaluations of the algorithms have been carried out in polluted megacities in developing country, associated with high density of buildings and heavy
anthropogenic pollutants. Nevertheless, the surface roughness and high aerosol loading in the boundary layer result in more complex structure and increase the difficulty of BLH retrieval based on these algorithms.

As one of the largest megacities in Asia affected by heavy pollution, Beijing provides a particular challenge to resolve the BLH determination. Actually, the movement of the atmosphere can affect the distribution of pollutant concentrations, and moreover, vertically propagating gravity waves influence the structure of the atmosphere and cause
some of the spatio-temporal variability (Fritts and Alexander, 2003). The gravity waves provide thereby new theory insights for the development of new algorithm in determining BLH by taking into account a probably insufficient vertical mixing of pollutants under weak thermal convection, and pollutant accumulation at high altitude due to long range transport process. Beijing, often governed by stagnant meteorological conditions, is surrounded by Mountains at the west, north and northeast, and characterized by favorable conditions to generate and maintain gravity waves. Such specific
atmospheric condition provides the opportunity to make insights into the difficulties related to the BLH retrieval based on existing algorithms, and to evaluate the performance of a new approach that considers the impact of gravity waves. Based on intensive observation campaign over Beijing, this paper aims at delving into the limitations of current retrieval algorithms employed for BLH determination from Lidar under polluted period, and coming up with the development of a new algorithm compatible with all atmospheric pollution conditions. This work therefore provides, for the first time, a
prototype of how to integrate into the BLH retrieval process, the gravity waves and the resulting complexity of low troposphere structure under heavy aerosol loading condition. Section 2 presents detailed description of Lidar observational experiment setting over Beijing and discusses the limitations of current algorithms for BLH retrieval; section 3 discusses the development of a new algorithm; section 4 presents the comprehensive evaluation of the new retrieval algorithm and comparative analysis with existing methods; conclusion and environmental implications are given
in section 5.

**2-Lidar Experiment Setting over Beijing and Evaluation of Existing BLH Retrieval Algorithms**



### 2.1 Lidar Observation Campaign

Beijing, the capital of China, is located at 39°56′N and 116°20′E on the northwest border of the Great North China Plain. It is surrounded by the Yanshan Mountains at the west, north and northeast. The topography favors accumulation of pollutants. The air pollution is critically high, with the peak concentration of $PM_{2.5}$ exceeding 500 μg/m³ (Sun et al.,

2014). An intensive observation campaign was conducted from Jul. 1 to Sep. 16, 2008, at the Institute of Atmospheric Physics, Chinese Academy of Sciences (39°58′28″N, 116°22′16″E), located between the north 3rd and 4th ring roads in Beijing and considered as a highly polluted urban site. A dual-wavelength (1064 nm, 532 nm) depolarization Lidar developed by the National Institute for Environmental Studies, Japan, sets on the rooftop of a 28 m-height building. The Lidar is used to retrieve the 6-m space-resolved and 10 s time-resolved aerosol vertical structure, but only for altitude >

100 m due to an incomplete overlap between the field of telescope view and the laser beam. More details of the Lidar parameters can be found in the research of Sugimoto et al. (2002) and Yang et al. (2010). $PM_{2.5}$ sampling was continuously conducted near the Lidar site. Potential temperature and relative humidity observed by radiosonde are integrated into the classic methods to retrieve the BLH and are employed to evaluate new algorithms (Stull, 1988).

Unprecedented 78 days intensive radiosonde campaign was conducted over the Institute of Atmospheric Physics site

(four times per day, 02:00, 08:00, 14:00 and 20:00 Local Standard Time) in line with the Lidar campaign at a radiosonde observatory located in southern of Beijing (39°48′N, 116°28′E). Daily $PM_{2.5}$ concentrations observed over the Institute of Atmospheric Physics site between July and September 2008 are shown in Fig. 1. A typical extended polluted episode occurred between Jul. 24 and 27, when the $PM_{2.5}$ concentration exceeded the Grade III National Ambient Air Quality Standards (Moderate pollution, GB3095-2012, 115 μg/m³, 24 h average). Jul. 24, 27 and 28 were respectively beginning

day, undergoing and ending of the pollution episode. Jul. 27 was the heaviest pollution day during the campaign, with $PM_{2.5}$ concentration of 195 μg/m³. On Jul. 27, the southeastern edge of a low-pressure system of North China prevailed over Beijing, inducing southerly flows. Under such meteorological condition, accumulation of pollutants (due to long range transport from neighboring regions) occurs over the south area (Chan and Yao, 2008). Below 850 hPa, warm advection over northern China triggers significant increase of air temperature at low altitudes, preventing the vertical

diffusion of pollutants (Fig. S1). This presents a typical condition for evaluating the performance of existing retrieval algorithms in determining the BLH.

### 2.2 Existing Gradient Algorithm for BLH Determination

In normal conditions of an aerosol-laden boundary layer and clean overlying free atmosphere, the gradient of the range-squared-corrected signal (RSCS) exhibits a strong negative peak at the transition between the boundary layer and free

atmosphere. Based on this principle, gradient algorithms were proposed and had become the most widely used ones. In this paper, we focus on the three most popular gradient methods including the first gradient method (GM), first logarithm gradient method (LGM) and normalized first gradient method (NGM). The optical power measured by Lidar is proportional to the signal backscattered of particles and molecules present in the atmosphere. The Lidar signal can be expressed by Eq. (1) below:

$$\mathrm{RS}(\lambda, r) = \frac{C}{r^2} E_0 \left[ \beta_m(\lambda, r) + \beta_p(\lambda, r) \right] T^2(\lambda, r) + RS_0 \tag{1}$$




where $\beta_p(\lambda,r)$ and $\beta_m(\lambda,r)$ are the particular and molecular backscatter coefficients, respectively, C is a constant for a given Lidar system, $E_0$ is the laser output energy, $T^2$ is the atmospheric transmission, $r$ is the range between the laser source and the target, $\lambda$ is the wavelength, and $RS_0$ is the background signal.

The RSCS is then defined in Eq. (2) by:

$$RSCS = (RS - RS_0)r^2 \qquad (2)$$

The first gradient method (GM), which assimilates the BLH to the altitude ($h_{GM}$) of the minimum gradient of the RSCS (Flamant et al., 1997;Hayden et al., 1997) is obtained by:

$$h_{GM} = \min[\frac{\partial RSCS}{\partial r}] \qquad (3)$$

The first logarithm gradient method (LGM) determines the BLH at the altitude, $h_{LGM}$, where the minimum of the first gradient of RSCS logarithm is reached (Senff et al., 1996). Such altitude is calculated by the equation:

$$h_{LGM} = \min[\frac{\partial \ln(RSCS)}{\partial r}] \qquad (4)$$

The normalized first gradient method (NGM) described below, estimates the BLH at the altitude where the normalized RSCS gradient reaches a minimum (He et al., 2006).

$$h_{NGM} = \min[\frac{\partial RSCS}{\partial r \times RSCS}] \qquad (5)$$

**2.3 Evaluation of Existing Algorithms Performance over Polluted Period**

As a key parameter for air pollution forecasting models, BLH can determine the volume available for the dispersion of pollutants (Seibert et al., 2000). Accurate retrieval of the BLH by automatic algorithms not only allows making insights into its diurnal fluctuations during pollution episodes, but also contributes to validate modeling results and improve prediction performance.

Prior to the calculation of the gradient with current three BLH retrieval algorithms, a moving average of 30 m in height was assumed in the stored Lidar profiles in accordance with the study of Pal et al. (2010) who previously reported that a height difference of 30 m was the most appropriate for identifying the minimum of the gradient. Typical gradient profiles of the RSCS and retrieved BLH from various algorithms with corresponding radiosonde profiles of the potential temperature and relative humidity are illustrated in Fig. 2b and Fig. 2c. Strong negative peaks were detected in the profiles for each algorithm to define the BLH (Fig. 2b). As illustrated in Fig. 2b, at 20:00 on Jul. 27, the BLH retrieved by GM is 480 m versus about 1590m retrieved by LGM and NGM. Determining the BLH from radiosonde measurements based on the potential temperature sharply increasing with altitude and decreasing relative humidity is the classic and most accurate approach usually applied to evaluate Lidar retrieval results (Seibert et al., 2000). At 20:00 on Jul. 27, the radiosonde identified a region at 1350 m, considered as actual BLH (Fig. 2c). Thus, GM significantly underestimated the BLH by approximately 870 m, while LGM and NGM overestimated the BLH by about 240 m. The diurnal cycle of the BLH retrieved by these algorithms is illustrated in Fig. 2a in comparison with the 4 radiosonde measurements (02:00, 08:00, 14:00, 20:00). The results demonstrated that none of the algorithms was able to fully capture the diurnal cycle of the BLH. The average underestimation was 500-600 m for the GM algorithm (strongly supporting previous finding of (He



et al., 2006)), against an overestimation of 400-500 m for the LGM and NGM algorithms on Jul. 27, in agreement with the profile analyses (Fig. 2b). In addition, the performance of the retrieval algorithms on July 24 and 28 (Fig. S2 and S3) strongly correlated with that found on Jul. 27. This highlights the critical bias and limitations of these algorithms in accurately determining the BLH under heavy aerosol loading.

**2.4 Limitation Analysis**

The top of boundary layer is often associated with strong gradients in the aerosol content, so that a simple negative gradient peak seems suitable to determine the BLH. However, data interpretation from aerosol Lidar is often not straightforward. Aerosol loading in low troposphere mainly originates from the ground level. Thus, under stable conditions, large negative gradient peaks possibly exist at near ground level (even larger than that of the BLH) due to insufficient vertical mixing of the pollutants in the boundary layer. Thus, the BLH might be wrongly determined by the GM based on these negative gradient peaks with critical underestimation. On the other hand, both LGM and NGM originally developed to filter out the influence of aerosols near the surface and to improve the original GM (Sicard et al., 2006; Emeis et al., 2007), result in an overestimation of the BLH. LGM is normally supposed to filter out the negative gradient peak near the ground to a certain extent, producing a higher BLH than GM (He et al., 2006). Such overestimation is probably induced by accumulation of aerosol at higher altitude due to adventive chemical transport (Stettler and Hoyningen-Huene, 1996), undetectable by the retrieval algorithms due to the impact of gravity waves on the atmosphere structure (Gardner, 1996), that inhibits the filtration skills of LGM and NGM. In clear, the accuracy of current retrieval algorithms in determining the BLH from Lidar is limited by heavy aerosol loading condition (with insufficient vertical mixing in the boundary layer) associated with the impact of vertically propagating gravity waves.

**3 Development of a New Algorithm**

**3.1 Rationale and Scientific Basis**

As evoked in previous Sections, heavy pollution and propagation of gravity waves critically limit the accuracy of current retrieval algorithm in determining the BLH from Lidar. Beijing is characterized by favorable conditions to generate and maintain gravity waves in particular due the presence of Qinghai-Tibet Plateau in the west, which is considered as potential source of gravity waves in Beijing (Gong et al., 2013). In fact, during more than two years campaign (from April 2010 to September 2011), daily and seasonal vertical mixing of wavelengths and phase velocities of 162 quasi-monochromatic gravity waves were observed over Beijing from Lidar (Gong et al., 2013). Moreover, statistical analysis of the captioned campaign revealed that gravity waves were maximal in summer (June-August), corresponding practically to discussed observation period of the present study (1 July-15 September). In clear, such finding serves as potential observational evidence of gravity wave and strong support of the present study. According to the research of Global Atmospheric Sampling Program, the gravity waves generated by the mountains are ~2-3 times higher than those generated by plains and oceans and ~ 5 times higher than those from other sources (Fritts and Alexander, 2003). Heavy air pollution episodes frequently occur in Beijing with stagnant meteorological conditions that maintain the gravity waves (Gibert et al., 2011).

The linear instability theory (LIT) of gravity waves (Dewan and Good, 1986) illustrated in Fig. 3. $m_b$ (buoyancy wave number), makes the transition between waves and turbulence (Gardner, 1996). Under $m > m_b$ condition, wind



fluctuations are dominated by turbulence, while under $m < m_b$, the fluctuations are governed by waves. The upper boundary layer is the transition between the boundary layer (where turbulence is the predominant process) and the free atmosphere (where large-scale waves can propagate vertically). The BLH is associated with $m_b$ to some extent. According to the research of Gardner et al. (1996), $F_u(m_b)$ (the spectrum of horizontal wind fluctuations) is proportional to $m_b^{-3}$ when

$m_b$ occurs as shown in Fig. 3 and by Eq. (6)).

$$m_b \propto F_u(m_b)^{-1/3} \tag{6}$$

Due to the dispersion relationship between the velocity and temperature fluctuations of gravity waves, $F_T(m_b)$ (the spectra of the fractional temperature) is proportional to the corresponding spectra of the horizontal velocity $F_u(m_b)$(Wang et al., 2000), Eq.(7) :

$$F_T(m_b) \propto F_u(m_b) \tag{7}$$

thus, $F_T(m_b)$ is also proportional to $m_b^{-3}$ when $m_b$ occurs as described in Eq.(8):

$$m_b \propto F_T(m_b)^{-1/3} \tag{8}$$

The ideal gas law can be written as Eq. (9):

$$P = \frac{1}{V} nRT \tag{9}$$

where P is the pressure of the gas, V is the volume of the gas, n is the amount of gas (in moles), R is the gas constant, and T is the absolute temperature of the gas. $n$ can be calculated by Eq. (10):

$$n = \frac{m}{\mu m_u} \tag{10}$$

where m is the mass of the gas mass, $m_u$ is the atomic mass constant, μ is the times of average molecular weight to $m_u$. Because $\rho = m/V$ (the density of the gas), Eq. (9) can be rewritten as Eq. (11):

$$P = \frac{1}{V} \frac{m}{\mu m_u} RT = \frac{R}{\mu m_u} \rho T \tag{11}$$

When pressure is constant, Eq. (11) can be rewritten as Eq. (12):

$$\frac{\partial \rho}{\rho} + \frac{\partial T}{T} = 0 \tag{12}$$

Eq. (12) shows that the fractional density $F_\rho(m_b)$ is proportional to the fractional temperature $F_T(m_b)$ when pressure is constant. Thus, $F_\rho(m_b)$ is also proportional to $m_b^{-3}$ when $m_b$ occurs (Eq. (13)).

$$m_b \propto F_\rho(m_b)^{-1/3} \tag{13}$$

Thus, $\lambda_b$ (buoyancy wavelength) is proportional to $F_\rho(\lambda_b)^{1/3}$

$$\lambda_b \propto F_\rho(\lambda_b)^{1/3} \tag{14}$$

Such equation determines the basis of the development of the new algorithm.





### 3.2 Algorithm Description

The motion of aerosol in the boundary layer is determined by the background atmosphere (the aerosol particles move with the background atmosphere). Thus, the aerosols and the background share the same fractional fluctuation. Thereby, $\lambda_b$ is also proportional to $F_{\rho(aerosol)}(\lambda_b)^{1/3}$ (the fractional aerosol density), as illustrated by Eq. (15):

$$\lambda_b \propto F_{\rho(aerosol)}(\lambda_b)^{1/3}$$

(15)

Eq. (15) means that the cubic root of $F_{\rho(aerosol)}(\lambda_b)$ reflects $\lambda_b$, corresponding to the top of boundary layer. Eq. (15) highlights that cubic root reflects the relationship of the BLH with fluctuant characteristics of aerosols at the position of BLH. Therefore, the BLH can be determined by capturing the fluctuant characteristic of aerosols. RSCS is proportional to $\rho_{Aerosol}$ (the density of aerosols) in accordance with the Fernald inversion algorithm of the aerosol Lidar equation (Fernald, 1984). The cubic root of the RSCS reflects the characteristics of $\lambda_b$ that corresponds to BLH; thus the cubic root of the RSCS can be applied to estimate the BLH as described in Eq (16).

The cubic root gradient method (CRGM), a new algorithm for BLH determination is thus defined by:

$$h_{CRGM} = \min[\frac{\partial(RSCS^{1/3})}{\partial r}]$$

(16)

With such new algorithm, the BLH corresponds to the altitude where the cubic root RSCS gradient reaches a minimum. This integrates the impact of gravity waves on the atmospheric structure in determining the BLH.

### 4 Evaluation of the New Algorithm and Comparative Analysis with Existing Methods

### 4.1 Under Heavy Polluted Episodes

Fig. 4b (similar to Fig. 3b) shows the BLH retrieved by CRGM in red dotted line. Strong negative peaks were detected in the profiles for each algorithm to define the BLH (Fig. 4b). At 20:00 on Jul. 27, the BLH retrieved by CRGM was 1350 m, in perfect agreement with the actual BLH determined by radiosonde (1350 m), against 480 m and 1590 m determined by LGM and NGM respectively. The diurnal cycles of the BLH retrieved by CRGM presented on Fig. 4a, show CRGM good capture of the unimodal diurnal cycle of the BLH, presenting a peak at 14:00-15:00 and a valley at 07:00-08:00, induced by the thermal activity of the ground. In comparison with CRGM, the BLH determined by LGM and NGM did not present unimodal diurnal cycles. On the other hand, although the GM-retrieved BLH showed unimodal diurnal cycle, the amplitudes of the valley and peak were lower. Comparing the 4-moment radiosonde-retrieved BLH (02:00, 08:00, 14:00, 20:00) with the algorithms results highlights that CRGM presents the least bias while GM shows an average underestimation of 500-600 m, and LGM and NGM result in an average overestimation of 400-500 m. Such performance of the retrieval algorithms is confirmed on Jul. 24 and Jul. 28 (Fig. S4 and Fig. S5).

In order to further compare the performance of the CRGM with the current algorithms on heavy polluted episodes, period of daily $PM_{2.5}$ concentrations exceeding the Grade II National Ambient Air Quality Standards (light pollution, GB3095-2012, 75 μg/m³, 24 h average) is particularly analyzed (in total 24 days). Corresponding four times radiosonde data are used to evaluate retrieval algorithms results. Cloudy and rainy weather conditions are ignored to prevent increase of bias during the BLH retrieval process BLH. There are 89 available samples for each algorithm. Fig.5 presents the discrepancies between the retrieval algorithm results and the radiosonde-detected BLH. Although the retrieval errors are





limited in CRGM, it shows slightly symmetric height bias distribution of about 200 m. In contrast, the LGM and NGM retrievals significantly overestimate the BLH, with the height bias range of 60-1110 m, exceeding 300 m for more than 85% of the measurements (orange region on Fig.5). The GM algorithm underestimates the BLH by 30-1140 m, with an underestimation of more than 300 m occurring on 70% of the time (purple region). CRGM accurately reproduces the

fluctuations of the BLH for these samples and shows the weakest bias in the BLH retrieval (Table 1). The CRGM algorithm also shows the best correlation, with a coefficient of 0.91, and the weakest Root mean square error (RMSE) (142 m). The correlation coefficient is enhanced by at least 0.44, and the RMSE is reduced by more than 400 m in comparison with corresponding results of other three algorithms. However, minor discrepancies still subsist between CRGM and radiosonde for some possible reasons: (1) the locations of the Lidar and radiosonde measurements are 24 km

apart, which may induce slight BLH bias; and (2) the potential temperature profile observed by radiosonde and the aerosol concentration profile observed by Lidar might be inconsistent as previously reported by (Hennemuth and Lammert, 2006). Nevertheless, CRGM shows the best performance in determining the BLH during pollution episodes induced by stagnant air.

### 4.2 Under Clean Atmosphere

To investigate the performance of the algorithms under clean meteorological conditions, comparison between the new and current algorithms is performed on Aug. 9, 2008 with low $PM_{2.5}$ concentrations (~81 μg/m³). Fig. 6a shows the boundary layer evolution in terms of the time-height cross-section observation of the background-subtracted and RSCS in the 532-nm channel collected on Aug. 9, 2008. Similar to the result noticed on Jul.27, the BLH presents an obvious diurnal cycle, with a valley at 08:00-09:00 and peak at 14:00-15:00, due to the thermal diurnal cycle of the ground surface. In contrast to

Jul. 27, the boundary increases more quickly with a rate of about 240 m/h from 09:00 to 15:00, corresponding to 1.7 time the rate on the polluted day. The maximum height of the boundary layer of approximately 1800 m is reached at 15:00, ~500 m higher than the height on the polluted day. Such difference between the BLH on polluted and clean days might be explained by the fact that heavy aerosol loading affects radiative forcing and lower the BLH on polluted days (Quan et al., 2013).

Fig. 6b shows strong negative peaks in the profiles for each algorithm to determine the BLH. At 14:00 on Aug. 9, the retrieved BLH for all algorithms is 1680 m, in perfect agreement with the actual BLH determined by radiosonde (1680 m). All diurnal cycle results converge at 14:00 on Aug. 9, demonstrating that all retrieval algorithms capture the overall diurnal cycle of the actual BLH. Such good performance of all the algorithms under clean meteorological conditions is a result of the homogenous vertical distribution of aerosols, since under clean conditions, mixing of the aerosols by strong

thermal convection is more sufficient due to weak pollutant loading. In addition, there is no obvious large negative gradient peak to disturb the determination of the BLH.

### 4.3 Under Various Pollution Levels

Under various air pollution conditions (all pollution levels), a total of 298 radiosondes measurements are analyzed to estimate the BLH with comparison to retrieval algorithms. Cases of nocturnal BLH below the useful Lidar signal (before

the overlap reaches 1) or thin cumulus clouds formation at upper boundary layer (resulting in large error in the retrieval), are neglected. The 298 samples are categorized into five groups according to the corresponding air pollution level (GB3095-2012). We compare retrieved BLH by the algorithms from Lidar with the radiosonde results in each group. As



illustrated in Fig. 7, the retrieval results of the CRGM are close to the 1:1 line, while the GM, LGM and NGM present large biases. The GM results are generally below the 1:1 line, highlighting an underestimation of the BLH. LGM and NGM in general overestimate the BLH in all five comparison groups. The RMSE ranges over 124-137 m for CRGM, against 124-642 m for the other three algorithms. Furthermore, the RMSE of the three existing algorithms increases with

the PM$_{2.5}$ concentrations (Table 2). For the GM algorithm, the RMSE increases from 124 to 629 m with an increase of PM$_{2.5}$ concentration from 35 to 250 μg/m$^3$. Similarly, the RMSE of LGM and NGM increases from ~130 to 540 m with the PM$_{2.5}$ concentration increase from 35 to 250 μg/m$^3$. High aerosol loading is therefore always associated with higher RMSE. In contrast, the RMSE of CRGM remains relatively constant with the changes of air pollution level. These results perfectly corroborate the findings discussed in sections 4.1 and 4.2. In clear, existing retrieval algorithms are only suitable

to the aerosol profiles similar to the "textbook" boundary layer development, while CRGM appears to be a robust technique for BLH determination by Lidar.

**5 Conclusions and Environmental Implication**

Lidar is an appropriate instrument to determine the boundary layer height with high temporal and vertical resolution. In this paper, Lidar intensive observation campaign was conducted in Beijing to thoroughly evaluate the limitations of the

current method for boundary layer height determination and develop an algorithm suitable to all pollution conditions. Incontestably, current commonly employed retrieval algorithms (first gradient method, logarithm gradient method, and normalized gradient method) are unable to determine the boundary layer height during heavy polluted episodes due to inhomogeneous vertical distribution of aerosols under stable meteorological conditions associated with the impact of vertically propagating gravity waves on the tropospheric structure. The gradient algorithm critically underestimates the

boundary layer height by 30-1140 m, with an underestimation higher than 300 m, occurring 70% of the time. The logarithm and normalized gradient methods overestimate the boundary layer height, exceeding 300 m for more than 85% of the time.

The newly developed method (the cubic root gradient) considers the linear instability theory of gravity waves to determine the boundary layer height by capturing the vertical movement of aerosol at the transition between waves and

turbulence. As a result, the cubic root gradient method describes the fluctuation of the boundary layer with best correlation (R$^2$ = 0.91) and the weakest RMSE (142 m) under various atmospheric pollution conditions. In comparison with current gradient methods, the new technique reduces the RMSE by 400 m minimum under all pollution conditions. The RMSE of existing retrieval algorithms typically varies with aerosol loading (high RMSE is always associated with heavy aerosol loading, and weak RMSE correlates with weak aerosol loading) while the RMSE of the new method

remains almost constant with the changes of air pollution levels. The cubic root gradient method appears therefore to be a robust technique for boundary layer height determination from Lidar.

In terms of environmental implication, such innovation would technically contribute to improve the accuracy of regionally spatio-temporal distribution models and forecasts of aerosol loadings for an effective pollution control measure, in particular over number of megacities in China, since accurately determining the boundary layer is one of important

factors of uncertainties and bias reduction for reasonable air pollution modeling and forecasts. However, further development and expansion of Lidar observation system are needed notably under cloudy and rainy conditions in order to provide with greater benefit to pollution control management.





*Acknowledgments.* This work was supported by the Natural National Science Foundation of China (NSFC) (41305115), the Commonweal Project of the Ministry of Environmental Protection (201409001), Program 863 (2014AA06AA06A512), and Program 973 (2014CB447900). Dr. Ting Yang is grateful for the invaluable emotional support received from her parents over years to overcome all the darkness periods, and the endless happiness received from her baby daughter.

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

40          **Table 1.** Statistical parameters for each Lidar retrieval algorithm compared with radiosonde measurements

|  | CRGM | GM | LGM | NGM |
|---|---|---|---|---|
| **Correlation Coefficient ($R^2$)** | 0.91 | 0.71 | 0.50 | 0.44 |
| **RMSE (m)** | 142 | 384 | 434 | 498 |




**Table 2.** Root mean square error (RMSE) for each Lidar retrieval method compared with radiosonde measurements

| PM$_{2.5}$ (µg/m³) | CRGM (m) | GM (m) | LGM (m) | NGM (m) |
|---|---|---|---|---|
| **0-35** | 124 | 124 | 137 | 129 |
| **35-75** | 123 | 133 | 238 | 227 |
| **75-115** | 135 | 213 | 320 | 418 |
| **115-150** | 154 | 310 | 346 | 434 |
| **150-250** | 137 | 629 | 636 | 643 |

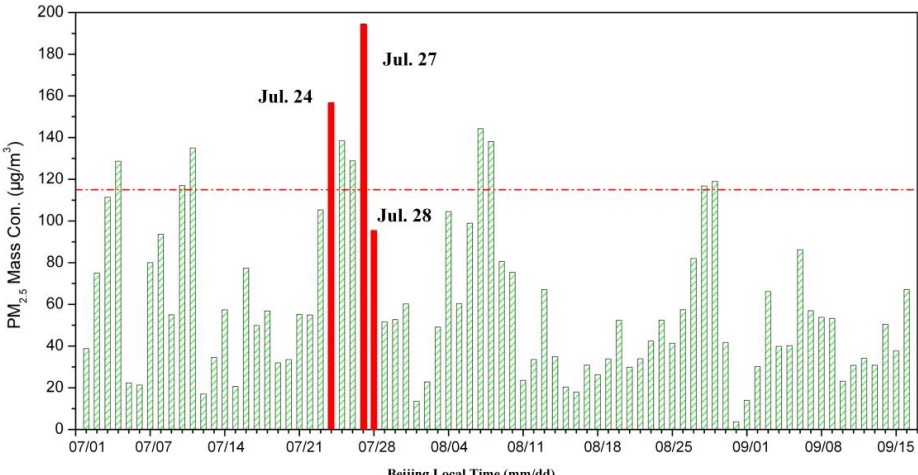

**Figure 1.** Daily variation of the PM$_{2.5}$ concentration between Jul. and Sep/ 2008 at IAP. The dotted line represents

10    the definition of a moderate pollution day (PM$_{2.5}$=115 µg/m³), and Jul. 24, 27 and 28 are highlighted.




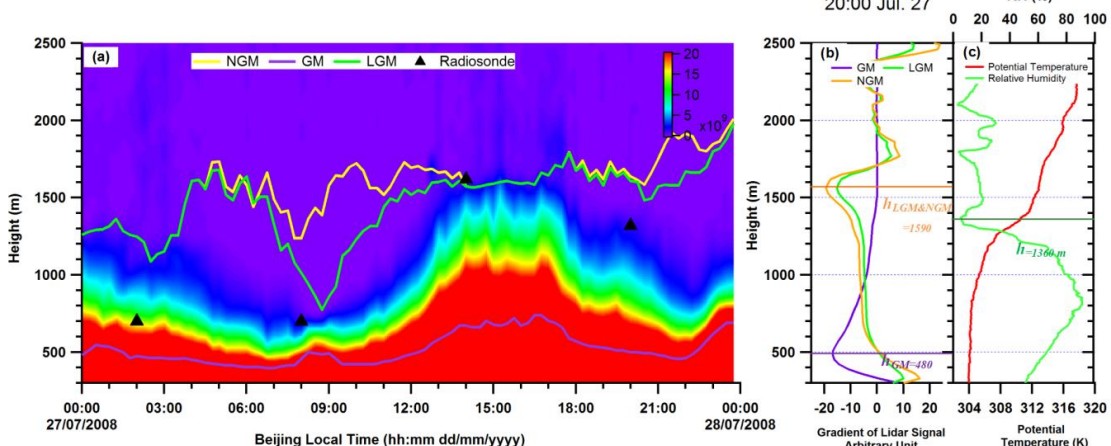

**Figure 2.** (a) Evolution of the Lidar range-squared-corrected signal (RSCS) at 532 nm on Jul. 27. The color scale indicates the intensity of the RSCS, and warm colors represent stronger light scattering. The diurnal BLH retrieved by LGM, GM and NGM are illustrated as green, purple and yellow lines, respectively. Black triangles show the BLH retrieved by radiosonde. (b) The profiles of LGM, GM and NGM, and the corresponding retrieval BLH at 20:00 on Jul. 27. LGM, GM and NGM are illustrated as green, purple and yellow lines, respectively, (c) Potential temperature and relative humidity at 20:00 on Jul. 27.

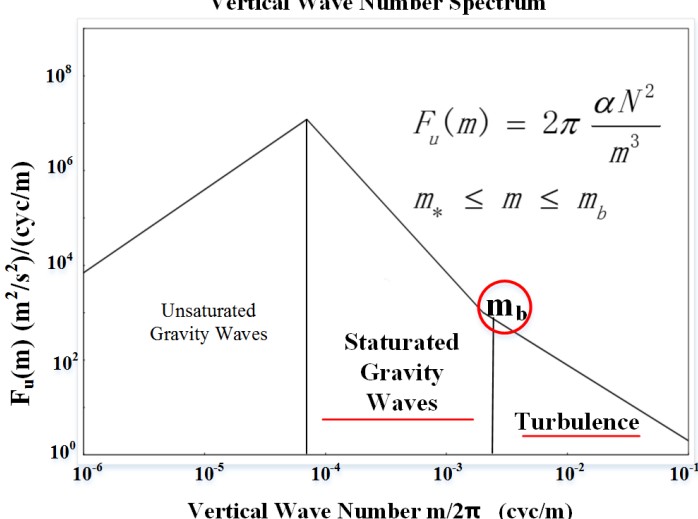

**Figure 3.** The canonical gravity wave vertical wave number spectrum of horizontal wind fluctuations. [a]

[a] From J. Atmos. Terr. Phys. **58,** 1577 (1996)





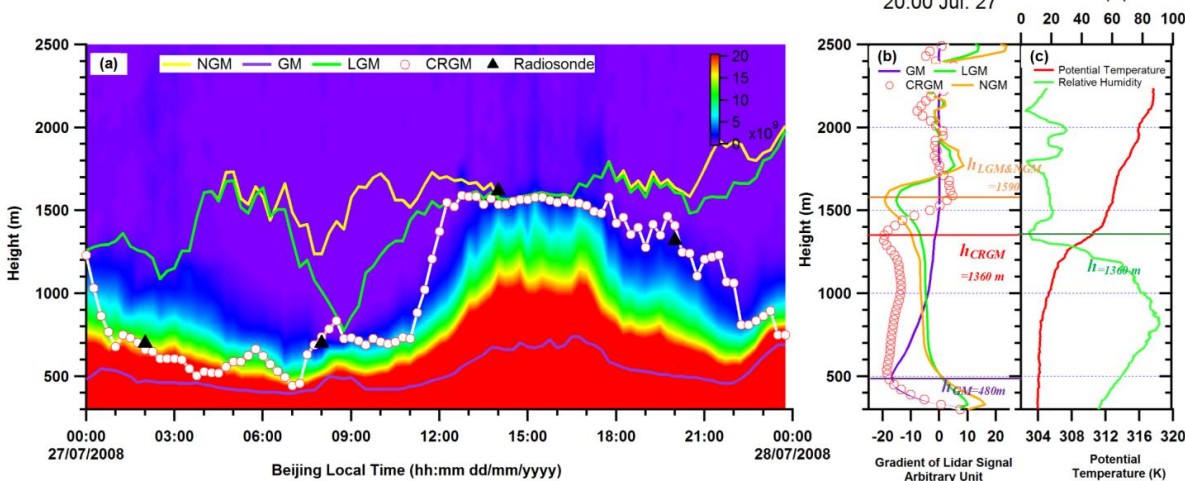

**Figure 4.** (a) Same as Fig. 2a with the addition of the diurnal BLH retrieved by CRGM as a white line with red outline and white circles. (b) Same as Fig. 2b with the addition of the CRGM profile as a red dotted line; (c) Same as Fig. 2c.

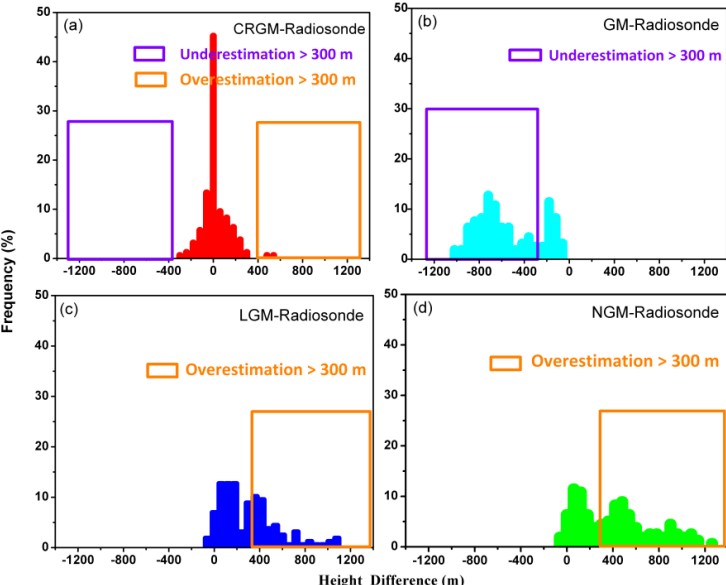

**Figure 5**. Histograms of the differences in BLH detected by radiosonde and by CRGM, NGM, LGM and GM. The X-axis is the height difference (m) between the retrieved BLH and that from radiosonde, and the Y-axis is the frequency of occurrence (%). The orange and purple regions highlight height differences of more than ±300 m.





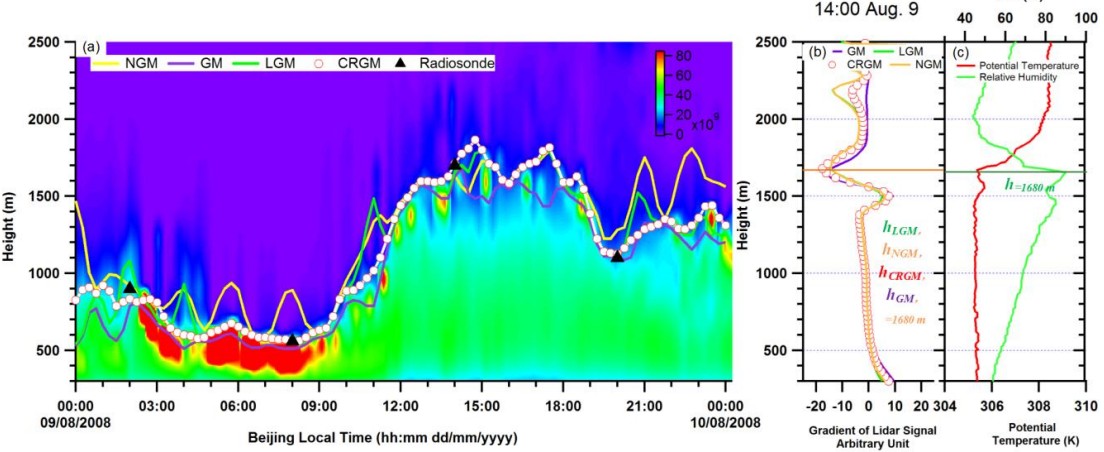

**Figure 6.** (a) Same as Fig. 4a but for Aug. 9; (b) Same as Fig. 4b but for 14:00 on Aug 9; (c) Same as Fig. 4c but for 14:00 on Aug 9.

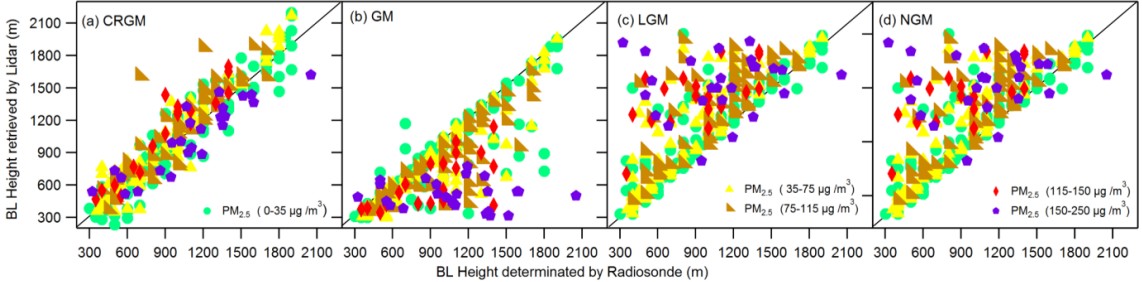

**Figure 7.** Comparison of $H_{CRGM}$, $H_{GM}$, $H_{LGM}$ and $H_{NGM}$ with the BLH retrieved from radiosonde measurements. The X-axis shows the radiosonde retrieval and the Y-axis is the Lidar retrieval using the different algorithms. The solid line indicates y=x; (a) CRGM, (b) GM, (c) LGM, (d) NGM. Different marks represent the comparisons under different pollution conditions (PM$_{2.5}$ concentrations). The comparisons under PM$_{2.5}$ concentrations less than 35 μg/m$^3$, 35-75 μg/m$^3$, 75-115 μg/m$^3$, 115-150 μg/m$^3$, and 150-250 μg/m$^3$ are shown as green circles, yellow triangles, brown triangles, red diamonds and purple hexagons, respectively.