# Peer review of "Technical note: Boundary layer height determination from Lidar for improving air pollution episode modelling: development of new algorithm and evaluation"

_Atmospheric Chemistry and Physics, 2016_

## Referee Comment (RC1) · Anonymous Referee #1 · 19 Jan 2017

Review of "Boundary layer height determination from Lidar for improving air pollution episode modelling: development of new algorithm and evaluation" by Ting Yang et al.

The boundary layer height (BLH) is one of the most important parameters in studying atmospheric boundary-layer problems. Measuring and determining the BLH are crucially important for predicting vertical exchange and transportation of air pollution. Lidar can be used to monitor the temporal and spatial variations of BLH over urban areas. However, the existing BLH retrieval algorithms are still limited due to the complexity of atmosphere conditions. This study presents a new algorithm based on gravity

wave theory to determine the BLH from Lidar and intercompare it with three existing algorithms. The result shows that the new algorithm has better performance than other three, especially for polluted episodes. This study will contribute to improve the accuracy of air quality modelling and forecasting systems. The quality of this manuscript can be improved (as detailed below). I recommend the publication of this paper in ACP after the following comments are properly addressed.

Major comments (1) The authors stated that the existing retrieval algorithms are only suitable to deal with the aerosol profiles like the "textbook" boundary layer development and the new one can be considered as a robust technique for BLH determination by Lidar. However, all the samples of aerosol vertical distribution presented in this manuscript are monotonous decrease of RSCS with increase altitude. In fact, the multi-layer aerosol structure is common in the ambient atmosphere, especially for polluted episodes when the long-distance transportation of aerosol pollutants is suspended in the low troposphere and entrained into PBL by vertical mixture process. In addition, the residual layer will usually appear in afternoon till midnight due to lacking turbulence driving force from ground heating, which will also entangle the aerosol vertical distribution. This multi-layer aerosol structure is just the difficulty and emphasis for BLH retrieval from Lidar profiles, just like the impact of cloud in BLH retrieval. It is better that the authors show some more cases of retrieval result for the multi-layer aerosol to verify the availability of new algorithm. A case shows that the new algorithm seems fail to distinguish residual layer from boundary layer from about 18:00-21:00 in Jul 24, 2008 (Figure S4). More discussions should be done to analyze the reason of fault retrieval result. (2) The three algorithms used for comparison are all "Gradient" based methods, the core of which is to find 'local' largest gradient, therefore, their performance are more likely to be impact by data's vertical resolution compare to algorithms focused more on the entire RSCS profile, such as ideal curve fit (Steyn et al. 2000) and wavelet method (Davis et al. 2000). I suggest the authors to do comparisons between CRGM and such algorithms, at least for several cases, to enrich the proof of CRGM's reliability.

Minor comments 1. P2, L4. "Duo to" should be "Due to". 2. P2, L24. "BHL" should be "BLH". 3. P7, L22. More states about cloud and rain mask method should be done in Section 2.

---

## Referee Comment (RC2) · Anonymous Referee #2 · 19 Jan 2017

This paper examined the existing algorithms of the BLH determination and found some of their persistent critical limitations, particularly over polluted episodes under weak thermal convection with high aerosol loading. They hardly can fully capture the diurnal cycle of the BLH due to pollutant insufficient vertical mixing in the boundary layer. A new approachïijŇthe cubic root gradient method (CRGM)ïijŇis proposed to overcome such weakness and to accurately reproduce the fluctuations of the BLH under various atmospheric pollution conditions. In comparison with the existing retrieval algorithms, the CRGM significantly reduced uncertainties in the BLH determination, largely increased the correlation coefficient from 0.44 to 0.91 and decreased the root mean
square error from 643 m to 142 m. The proposed method likely contributes to improve the accuracy of the BLH determination and air quality modelling and forecasting systems. The results are of interests to the community. Personally, I recommend it to be published. However, I also have some concerns about the rationale and scientific basis of the CRGM approach. The authors put their cubic root gradient method in an analog to the cubic law of gravity wave-turbulence spectrum. This sound plausible, provided that two major issues can be addressed in more details.

(1) How the gravity wave hypothesis can explain the diurnal cycle of the BLH? Is this due to tidal effects similar to that of gravity waves, or due to tidal modulation of gravity waves through their interaction, or both?

(2) As mentioned in the paper, the topography and meteorological conditions were favorable for the generation of gravity waves. Is there any observational evidence showing the existence of the gravity waves and strong vertical mixing associated with the waves during the discussed observation period? Such evidence would be a strong support to the hypothesis.

Some minor corrections are suggested in the commented pdf file.

**ACPD**

---

## Referee Comment (RC3) · Anonymous Referee #3 · 31 Jan 2017

General comment:

This manuscript presents "Boundary layer height determination from Lidar for improving air pollution episode modelling: development of new algorithm and evaluation". The authors discuss the boundary layer height under different pollution situations by using a new algorithm. As we know it is not easy to define the boundary height particularly, the data form radio sounding is usually too rough to define it. This paper provides a new thinking by using high resolution of lidar monitoring data to estimate boundary layer height. However, here still have some concerns in the following as list in specific

comments.

Specific comments:

1. Since the existing of gravity wave is an important factor for the method, should you declare the application limitation of the method you developed?

2. In general, the data from radio sounding is usually too rough to define boundary height. You might also need to declare what kind of the method you define from radiosondes (same as Stull (1988) ?), and in case have multi-levels, how do you define boundary layer height ?

3. Page 8, line 33 and Table 2, there are 298 radiosondes in total for the comparison of RMSE and Lidar retrieval method, how many cases in each pollution types? I suggest you need to show in the table.

4. Figure 6, from the case under clean atmosphere (9 Aug.), they are with good performance for all the method of retrieval algorithms, even in the vertical distribution. However, in Figure 7, at low concentration level (green dots, less than 35 ug/m3), the scatter distribution of some cases are diverse (i.e. far away 1:1 line) in different method. Why?

---

## Author Comment (AC1) · 7 Apr 2017

**Reply to Referee #1**

The authors would like to thank the reviewer for their constructive and valuable comments. Accordingly, point-by-point answer is given as follows:

*(1)  The three algorithms used for comparison are all "Gradient" based methods, the core of which is to find 'local' largest gradient, therefore, their performance is more likely to be impact by data's vertical resolution compare to algorithms focused more on the entire RSCS profile, such as ideal curve fit (Steyn et al. 2000) and wavelet method (Davis et al. 2000). I suggest the authors to do comparisons between CRGM and such algorithms, at least for several cases, to enrich the proof of CRGM's reliability.*

Following the reviewer's suggestion, the performance of the CRGM and existing other gradient methods in retrieving the boundary layer height (BLH) has been compared against the ideal curve fit and wavelet methods under five cases including starting, undergoing, ending of a heavy polluted episode, clean atmosphere and multi-layer condition. In ideal curve fit method, entrainment zone thickness (EZT) is set to be 2.77$s$ (Steyn et al. 1999) while Haar function with a median dilation of 150 m is employed in wavelet method (Davis et al. 2000). Detailed discussion case by case is given below.

**-Undergoing heavy polluted episode (27 July)**

As illustrated in ReFig.1-1b, at 20:00 on 27 July (heavy polluted episode), the BLH retrieved by CRGM is 1350 m, in good agreement with radiosonde result (1350 m), against 720m for ideal curve fit and a maximum of 540m for the wavelet covariance transform coefficient (ReFig.1-1c, d). Obviously, ideal curve fit and wavelet methods significantly underestimate the BLH by approximately 630 m and 810 m, respectively, and seem to be unable to fully capture the diurnal cycle of the BLH as showed on ReFig.1-1a. Such underestimate of the BLH from the wavelet method was also previously reported by several studies (Sawyer and Li, 2013; Wang et al., 2012; Su et al., 2017).

[Figure]

**ReFig.1-1**: (a) Evolution of the Lidar range-squared -corrected signal (RSCS) at 532 nm on 27Jul. The color bar indicates the intensity of the RSCS; the diurnal BLH retrieved by CRGM, three typical gradient method (LGM, GM and NGM), ideal curve fit and wavelet methods are illustrated as red dotted line, green, purple, yellow, origin and brown lines, respectively; (b) the profile of CRGM and the other three gradient methods and the corresponding retrieval BLH at 20:00 27 July, CRGM is illustrated in red dotted line and LGM, GM and NGM are illustrated as green, purple and yellow lines, receptively; (c) RSCS signal in red line, and ideal fit curve in green line; (d) RSCS signal in red line and wavelet covariance transform (WCT) in brown line; (e) Potential temperature and relative humidity at 20:00 on 27 July.

**-Starting heavy polluted episode (24 July)**

The diurnal cycle of BLH retrieved by these algorithms in a starting day of heavy polluted episode and typical profiles of the methods are illustrated in ReFig.1-2. The results are similar with those found for undergoing polluted episode. The BLH

retrievals from ideal fit curve and wavelet methods are significantly underestimation with a maximum of ~600 m and ~ 800 m, in particular in noon and afternoon.

[Figure]

**ReFig.1-2**: same as ReFig.1-1, but for 24 July.

**-Ending heavy polluted episode (28 July)**

The comparison results are showed on ReFig.1-3 and confirmed persistent underestimation of BLH retrievals from the ideal fit curve and wavelet methods (with a maximum underestimation of ~700 m and ~1000 m respectively), more obvious in moon and afternoon (as found in previous cases).

[Figure]

**ReFig.1-3**: same as ReFig.1-1, but for 28 July.

**-Clean atmosphere day (9 Aug)**

In this case, the results show that all retrieval algorithms capture relatively well the overall diurnal cycle of the BLH (ReFig.1-4). However, the retrieval from the ideal curve fit seems to present a slight underestimation of 150m at 14:00 9 Aug (ReFig.1-4c).

[Figure]

**ReFig.1-4**: same as ReFig.1-1, but for 9 Aug.

**-Multi-layer condition**

Proper analysis of algorithm performance under multi-layer condition is given with the reply to comment (3).

*(2)  A case shows that the new algorithm seems fail to distinguish residual layer from boundary layer from about 18:00-21:00 in Jul 24, 2008 (Figure S4). More discussions should be done to analyze the reason of fault retrieval result.*

Yes, exactly! The new algorithm fails to define the boundary layer height for 3 hours on 24 July. By further examining the mean profile of CRGM over 18:00-21:00, as illustrated in ReFig.1-5b, it presents two significant negative peaks (1680 m and 960 m). We believe that the boundary layer height determination might be significantly disturbed by the largest negative value (at 1680m), resulting in a failure of the CRGM.

[Figure]

**ReFig.1-5**: same as ReFig.1-1, (b) the main gradient profile of CRGM during 18:00-21:00 Jul24.

Actually, in our manuscript, the main purpose is to evaluate the performance of the algorithms (advantages and limitations) without further data quality control of the retrieval BLH, thus the BLH results provided in Fig.S4 are independent from moment to moment. However, when carefully considering the data quality checking process (usually applied in such circumstance) consisting of (1) Searching and checking for more accurate negative peak under BLH decreasing condition at sunset (if the BLH is determined by the largest negative peak, higher than that of previous moment), (2) eliminating the fault negative peaks according to BLH climatic condition at near ground level or high altitude (over 3 km in summer, >1 km at night time), (3) comparing with neighboring moment results and ensuring the retrieval results are reliable in time and space. By applying the above quality checking processes, the determination of the BLH without failure is relatively assured (960 m at second largest negative gradient peak in this case, ReFig.1-5b).

*(3) This multi-layer aerosol structure is just the difficulty and emphasis for BLH retrieval from Lidar profiles, just like the impact of cloud in BLH retrieval. It is better that the authors show some more cases of retrieval result for the multi-layer aerosol to verify the availability of new algorithm.*

In accordance with the reviewer suggestion, we have applied all algorithms to the most complex case for multi-layer aerosol structure in our database, occurring on 26 July (ReFig1-6), in order to provide further discussion on the BLH retrieval

performance and limitation. Meanwhile, some improving suggestions also discussed for complex multi-layer cases.

As illustrated in ReFig.1-6a, an obvious two-layer aerosol structure is observed from 00:00-04:00 Jul.26, associated with a low aerosol concentration "hole" extended from ~300m to 1300m. The BLH at noon time (12:00) of the day before (Jul.25) is about 1500m, then gradually decreases to ~1300 m at 20:00 due to the ground cooling in absence of solar short-wave radiation. Significant variations of aerosol signal at upper and lower parts are also observed on 26 July between 00:00 and 04:00. Basically, the formed upper and lower parts seem not to be the natural residual and nocturnal boundary layer, but probably rather triggered by large wind driving force.

[Figure]

**ReFig.1-6**: same as ReFig.1-1, but for 26 July.

The BLH is defined at 1380m by CRGM, NGM, GM, LGM and the ideal fit curve, and at 1350 m by the wavelet transform method at 02:00 Jul.26 (ReFig.1-6 b, c ,d). However, all algorithms seem to only track the top of the residual layer over the multi-layer sequence (00:00-04:00 Jul. 26) and fail to capture the nocturnal layer top.

Furthermore, analysis of the mean vertical profiles of the algorithms (ReFig.1-6 b, c ,d; ReFig.1-7b,c,d), enables to notice two obviously negative peaks induced by CRGM, NGM, GM, LGM and the wavelet method profiles (a first largest gradient peak noticeable at ~1380 m on the top of residual layer, and a second largest peak occurring at 510 m near the top of nocturnal boundary layer). Even though all algorithms fail to track the BLH variation induced by the first largest peak, the second largest peaks work well for nocturnal boundary layer with CRGM, CRGM, NGM, GM, LGM and the wavelet method when applying the data quality control checking as described in the reply to comment (2). The retrieved nocturnal boundary layer height variation by each algorithm with data control during the multi-layer period is illustrated in ReFig.1-7a.

[Figure]

**ReFig.1-7**: same as ReFig.1-6, but after data quality control.

*(4) P2, L4. "Duo to" should be "Due to".*

   It has been revised in the manuscript.

(5) *P2, L24. "BHL" should be "BLH".*

It has been revised in the manuscript.

*(6) More states about cloud and rain mask method should be done in Section 2.*

The cloud and rain detection follows the methods employed by Asian dust and aerosol Lidar observation network (AD-net) in East Asia, which was supported by world meteorological organization (WMO) Global Atmosphere Watch (GAW) program. Rain was detected by color ratio ($\gamma'$, the ratio of $\beta_{1064}'$ to $\beta_{532}'$) to distinguish rainy and clear (no rain) regions, in which, $\beta_{1064}'$ *and* $\beta_{532}'$ present the attenuated backscatter coefficient at 1064 nm and 532 nm, respectively. Large droplets have a large $\gamma'$ value, so once $\gamma'$ exceeds a threshold (1.1) over a certain vertical internal in the lower atmosphere, the profile is classified as a rain profile. Cloud base height is determined by the vertical gradient of $\beta_{1064}'$ and the peak value of $\beta_{1064}'$ between the cloud base and the apparent cloud top. The detailed description of the method is provided by Shimizu et al., (2016).

Davis, K. J., Gamage, N., Hagelberg, C. R., Kiemle, C., Lenschow, D. H., and Sullivan, P. P.: An Objective Method for Deriving Atmospheric Structure from Airborne Lidar Observations, J. Atmos. Oceanic Technol., 17, 1455-1468, 2000.

Sawyer, V., and Li, Z.: Detection, variations and intercomparison of the planetary boundary layer depth from radiosonde, lidar and infrared spectrometer, Atmos. Environ., 79, 518-528, 10.1016/j.atmosenv.2013.07.019, 2013.

Steyn, D. G., Baldi, M., and Hoff, R. M.: The Detection of Mixed Layer Depth and Entrainment Zone Thickness from Lidar Backscatter Profiles, J. Atmos. Oceanic Technol., 16, 953-959, 1999.

Shimizu, A., Nishizawa, T., Jin, Y., Kim, S.-W., Wang, Z., Batdorj, D., and Sugimoto, N.: Evolution of a lidar network for tropospheric aerosol detection in East Asia, OPTICE, 56, 031219-031219, 10.1117/1.OE.56.3.031219, 2016.

Su, T., Li, J., Li, C., Xiang, P., Lau, A. K.-H., Guo, J., Yang, D., and Miao, Y.: An intercomparison of long-term planetary boundary layer heights retrieved from CALIPSO, ground-based lidar and radiosonde measurements over Hong Kong, J. Geophys. Res:Atmos., 10.1002/2016jd025937, 2017.

Wang, Z., Cao, X., Zhang, L., Notholt, J., Zhou, B., Liu, R., and Zhang, B.: Lidar measurement of planetary boundary layer height and comparison with microwave profiling radiometer observation, Atmos. Meas. Tech., 5, 1965-1972, 10.5194/amt-5-1965-2012, 2012.

---

## Author Comment (AC2) · 7 Apr 2017

The authors would like to thank the editor and reviewers for their constructive and valuable comments. Accordingly, point-by-point answer is given as follows:

**Referee #2**

*(1)  How the gravity wave hypothesis can explain the diurnal cycle of the BLH? Is this due to tidal effects similar to that of gravity waves, or due to tidal modulation of gravity waves through their interaction, or both?*

Great thanks to the reviewer for this important question!

Basically, we believe that the interaction between gravity waves and the BLH might be similar to that between tide and gravity waves. Existing literature did not clearly focus on how the gravity wave hypothesis can explain the diurnal cycle of the BLH. However, concise review of the interactions between tide and gravity waves may help to understand the impact of gravity waves on the BLH diurnal cycle.

In fact, as reported by Williams et al. (1999), tide presents diurnal cycle in amplitude. Moreover, observational and modeling studies found that interactions between tide and gravity waves could result in significant modifications of the tidal diurnal cycle (Fritts and Vincent, 1987;Meyer, 1999). Momentum deposition by gravity wave could alter the background wind (by acceleration or deceleration), and significantly modulate the diurnal tidal structure (amplitude, phase and location of peaked amplitude and vertical wavelength etc.) as reported by Liu et al. (2013), Thayaparan et al. (1995), and England et al. (2006). Reciprocally, the evolution of gravity wave (breaking or saturating process) could also be profoundly modulated by the tidal wave (Liu et al., 2000;Preusse et al., 2001). In addition, eddy diffusion from breaking gravity waves was also found to have strong influence on tidal amplitudes (Meyer, 1999). Thus, by considering the fact that the BLH also presents diurnal cycle (same as tide), its interactions with gravity waves might plausibly be similar to those between tide and gravity waves.

England, S. L., Dobbin, A., Harris, M. J., Arnold, N. F., and Aylward, A. D.: A study into the effects of gravity wave activity on the diurnal tide and airglow emissions in the equatorial mesosphere and lower thermosphere using the Coupled Middle Atmosphere and Thermosphere (CMAT) general circulation model, J. Atmos. Sol.-Terr. Phy., 68, 293-308, http://dx.doi.org/10.1016/j.jastp.2005.05.006, 2006.

Fritts, D. C., and Vincent, R. A.: Mesospheric Momentum Flux Studies at Adelaide, Australia: Observations and a Gravity Wave–Tidal Interaction Model, Journal of the Atmospheric Sciences, 44, 605-619, 10.1175/1520-

0469(1987)044<0605:mmfsaa>2.0.co;2, 1987.

Liu, A. Z., Lu, X., and Franke, S. J.: Diurnal variation of gravity wave momentum flux and its forcing on the diurnal tide, J. Geophys. Res:Atmos., 118, 1668-1678, 10.1029/2012jd018653, 2013.

Liu, H.-L., Hagan, M. E., and Roble, R. G.: Local mean state changes due to gravity wave breaking modulated by the diurnal tide, J. Geophys. Res:Atmos., 105, 12381-12396, 10.1029/1999jd901163, 2000.

Meyer, C. K.: Gravity wave interactions with the diurnal propagating tide, J. Geophys. Res:Atmos., 104, 4223-4239, 10.1029/1998jd200089, 1999.

Preusse, P., Eckermann, S. D., Oberheide, J., Hagan, M. E., and Offermann, D.: Modulation of gravity waves by tides as seen in CRISTA temperatures, Adv. Space Res., 27, 1773-1778, http://dx.doi.org/10.1016/S0273-1177(01)00336-2, 2001.

Thayaparan, T., Hocking, W. K., and MacDougall, J.: Observational evidence of tidal/gravity wave interactions using the UWO 2 MHz radar, Geophys. Res. Lett., 22, 373-376, 10.1029/94GL03270, 1995.

Williams, P. J. S., Mitchell, N. J., Beard, A. G., Howells, V. S. C., and Muller, H. G.: The coupling of planetary waves, tides and gravity waves in the mesosphere and lower thermosphere, Adv. Space Res., 24, 1571-1576, http://dx.doi.org/10.1016/S0273-1177(99)00881-9, 1999.

*(2) As mentioned in the paper, the topography and meteorological conditions were favorable for the generation of gravity waves. Is there any observational evidence showing the existence of the gravity waves and strong vertical mixing associated with the waves during the discussed observation period? Such evidence would be a strong support to the hypothesis.*

Special thanks to the reviewer for such comment!

Actually, before officially disclosing the data for the present study, an intensive observation campaign of gravity wave evidence over Beijing was previously conducted from April 2010 to September 2011(Gong et al., 2013). Thus, during more than two years campaign, daily and seasonal vertical mixing wavelengths and phase velocities of 162 quasi-monochromatic gravity waves associated with the topography (impact of surrounding mountainous plateaus, in particular the Qinghai-Tibet Plateau) and meteorology were observed over Beijing from Lidar. Moreover, as reported by the authors, statistical analysis of the captioned campaign revealed that gravity waves were notably maximal in summer (June-August), corresponding practically to discussed observation period of the present study (1 July-15 September). In short, findings reported by Gong et al. (2013) serve as potential evidence of gravity wave and strong support of the present study on the development of new algorithm for BLH

determination from Lidar. The present complementary information has been integrated into the manuscript (Section, 3.1 Rationale and Scientific Basis).

**3 Development of a New Algorithm**

**3.1 Rationale and Scientific Basis**

As evoked in previous Sections, heavy pollution and propagation of gravity waves critically limit the accuracy of current retrieval algorithm in determining the BLH from Lidar. Beijing is characterized by favorable conditions to generate and maintain gravity waves in particular due the presence of Qinghai-Tibet Plateau in the west, which is considered as potential source of gravity waves in Beijing (Gong et al., 2013). In fact, during more than two years campaign (from April 2010 to September 2011), daily and seasonal vertical mixing of wavelengths and phase velocities of 162 quasi-monochromatic gravity waves were observed over Beijing from Lidar (Gong et al., 2013). Moreover, statistical analysis of the captioned campaign revealed that gravity waves were maximal in summer (June-August), corresponding practically to discussed observation period of the present study (1 July-15 September). In clear, such finding serves as potential observational evidence of gravity wave and strong support of the present study. According to the research of Global Atmospheric Sampling Program, the gravity waves generated by the mountains are ~2-3 times higher than those generated by plains and oceans and ~ 5 times higher than those from other sources (Fritts and Alexander, 2003). Heavy air pollution episodes frequently occur in Beijing with stagnant meteorological conditions that maintain the gravity waves (Gibert et al., 2011).

Fritts, D. C., and Alexander, M. J.: Gravity wave dynamics and effects in the middle atmosphere, Rev. Geophys., 41, 10.1029/2001rg000106, 2003.

Gong, S., Yang, G., Xu, J., Wang, J., Guan, S., Gong, W., and Fu, J.: Statistical characteristics of atmospheric gravity wave in the mesopause region observed with a sodium lidar at Beijing, China, J. Atmos. Sol.-Terr. Phy., 97, 143-151, 10.1016/j.jastp.2013.03.005, 2013.

Gibert, F., Arnault, N., Cuesta, J., Plougonven, R., and Flamant, P. H.: Internal gravity waves convectively forced in the atmospheric residual layer during the morning transition, Quart. J. Roy. Meteor. Soc., 137, 1610-1624, 10.1002/qj.836, 2011.

---

## Author Comment (AC3)

**Reply to Referee #3**

The authors would like to thank the editor and reviewers for their constructive and valuable comments. Accordingly, point-by-point answer is given as follows:

(1) ***Since the existing of gravity wave is an important factor for the method, should you declare the application limitation of the method you developed?***

The favorable conditions for generation and maintaining gravity waves over Beijing (Gong et al., 2013; Gibert et al., 2011) provide us with great opportunity to develop the new BLH retrieval algorithm based on long-term Lidar observations. Such new method is thus supposed to be effective for the BLH retrieval from Lidar under similar condition of gravity wave impact over other areas. However, this method is out of application under very shallow nocturnal boundary layer below the useful Lidar signal (before the overlap reaches 1), especially in winter night. In addition, more applications of the new algorithm under gravity wave impact conditions over various areas worldwide are necessary to properly evaluate its fluctuations and possibly further limitations in the future.

Gibert, F., Arnault, N., Cuesta, J., Plougonven, R., and Flamant, P. H.: Internal gravity waves convectively forced in the atmospheric residual layer during the morning transition, Quart. J. Roy. Meteor. Soc., 137, 1610-1624, 10.1002/qj.836, 2011.

Gong, S., Yang, G., Xu, J., Wang, J., Guan, S., Gong, W., and Fu, J.: Statistical characteristics of atmospheric gravity wave in the mesopause region observed with a sodium lidar at Beijing, China, J. Atmos. Sol.-Terr. Phy., 97, 143-151, 10.1016/j.jastp.2013.03.005, 2013.

*(2) In general, the data from radio sounding is usually too rough to define boundary height. You might also need to declare what kind of the method you define from radiosondes (same as Stull (1988)?), and in case have multi-levels, how do you define boundary layer height?*

The BLH has been defined from the radio sounding based on an elevated inversion in potential temperature (or the height of a significant reduction in air moisture), which is the classic and easy way, widely employed by Stull (1988), Seibert et al (2000) and He et al (2006). Accordingly, intensive radio sounding has been achieved 4 times per day with vertical resolution of ~ 7-10 m. An illustration of BLH determination based on such method is displayed on ReFig.3-1 (14:00 on 9 Aug).

[Figure]

ReFig.3-1 (a) potential temperature and relative humidity at 14:00 Aug.9 (b) the change of potential temperature and relative humidity with altitude

For the cases with multi-layers, the potential temperature presents multi-inversion associated with complex relative humidity profile. As such, wind and wind shear profile method could be applied to define the BLH where the wind shear first becomes less than a detection criterion (Hyun, et al.2005) as described below.

$$\sqrt{(\frac{\partial \overline{U}}{\partial z})^2 + (\frac{\partial \overline{V}}{\partial z})^2} < S_c \tag{2}$$

$\overline{U}$ and $\overline{V}$ are the mean wind components in the east-west and north-south directions, respectively; $Sc$ is the detection criterion, Usually, $Sc$ is set to be 0.04 s$^{-1}$.

He, Q. S., Mao, J. T., Chen, J. Y., and Hu, Y. Y.: Observational and modeling studies of urban atmospheric boundary-layer height and its evolution mechanisms, Atmos. Environ., 40, 1064-1077, http://dx.doi.org/10.1016/j.atmosenv.2005.11.016, 2006.

Seibert, P., Beyrich, F., Gryning, S.-E., Joffre, S., Rasmussen, A., and Tercier, P.: Review and intercomparison of operational methods for the determination of the mixing height, Atmos. Environ., 34, 1001-1027, http://dx.doi.org/10.1016/S1352-2310(99)00349-0, 2000.

Stull, R. B.: An introduction to boundary layer meteorology, Springer, 1988. P13

Hyun, Y.-K., Kim, K.-E., and Ha, K.-J.: A comparison of methods to estimate the height of stable boundary layer over a temperate grassland, Agricultural and Forest Meteorology, 132, 132-142, 10.1016/j.agrformet.2005.03.010, 2005.

*(3) Page 8, line 33 and Table 2, there are 298 radiosondes in total for the comparison of RMSE and Lidar retrieval method, how many cases in each pollution types? I suggest you need to show in the table.*

We have updated Table1 as following:

Table 1: Root-Mean-Square Error (RMSE) for each Lidar retrieval method compared with radiosonde measurements and Sample size in each comparison level

| PM$_{2.5}$ (µg/m$^3$) | CRGM (m) | GM (m) | LGM (m) | NGM (m) | Samples / Cases |
|---|---|---|---|---|---|
| **0-35** | 124 | 124 | 137 | 129 | 114 |
| **35-75** | 123 | 133 | 238 | 227 | 88 |
| **75-115** | 135 | 213 | 320 | 418 | 53 |
| **115-150** | 154 | 310 | 346 | 434 | 19 |
| **150-250** | 137 | 629 | 636 | 643 | 24 |

*(4) Figure 6, from the case under clean atmosphere (9 Aug.), they are with good performance for all the method of retrieval algorithms, even in the vertical distribution. However, in Figure 7, at low concentration level (green dots, less than 35 ug/m3), the scatter distribution of some cases are diverse (i.e. far away 1:1 line) in different method. Why?*

Basically, under low aerosol loading condition (clean condition), the RMSE of the retrieval algorithms is found to be relatively weak (good performance). However, persistent high bias might be noticeable under some exceptional conditions as pointed out by the reviewer. Actually, the algorithms result is determined by moment profile, and in clean condition, some occasional ground floating dust might disturb the moment signal of ideal vertical distribution, resulting in a bias of the algorithm retrieval result. ReFig.3-2 illustrates such a case with PM$_{2.5}$ concentration of 9.5µg/m$^3$ (clean condition), with large bias between BLH determined by radiosonde and Lidar algorithms. In clear, we believe that observed bias might be due to such occasional floating dust.

[Figure]

ReFig.3-2 (a) the Profiles of CRGM, LGM, GM and NGM, and the corresponding retrieval BLH at 02:00 Jul.5 (b) potential temperature and relative humidity at 02:00 Jul.5